# Acceptability of Telerehabilitation: Experiences and Perceptions by Individuals with Stroke and Caregivers in an Early Supported Discharge Program

**DOI:** 10.3390/healthcare12030365

**Published:** 2024-01-31

**Authors:** Fatimata Ouédraogo, Louis-Pierre Auger, Emmanuelle Moreau, Odile Côté, Rosalba Guerrera, Annie Rochette, Dahlia Kairy

**Affiliations:** 1School of Rehabilitation, Faculty of Medicine, Université de Montréal, Montréal, QC H3N 1X7, Canada; annie.rochette@umontreal.ca (A.R.); dahlia.kairy@umontreal.ca (D.K.); 2Centre for Interdisciplinary Research in Rehabilitation of Greater Montreal (CRIR), Montréal, QC H3S 1M9, Canada; louis-pierre.auger@umontreal.ca; 3Institute of Health Sciences Education, Faculty of Medicine, McGill University, Montréal, QC H3G 1A1, Canada; 4Institut Universitaire sur la Réadaptation en Déficience Physique de Montréal, Centre Intégré Universitaire de Santé et Services Sociaux (CIUSSS) du Centre-Sud-de-l’Île-de-Montréal, Montréal, QC H3S 2J4, Canada; emmanuelle.moreau@umontreal.ca (E.M.); odile.cote.ccsmtl@ssss.gouv.qc.ca (O.C.); rosalba.guerrera.ccsmtl@ssss.gouv.qc.ca (R.G.)

**Keywords:** telerehabilitation, stroke, technology, early supported discharge, home-based, Canada

## Abstract

Introduction: Telerehabilitation (TR) is a promising method for facilitating the delivery and access to post-stroke rehabilitation services. Objective: The aim of this study was to explore the acceptability of TR and factors influencing its adoption by individuals with stroke and caregivers. Methods: A qualitative descriptive approach was used. Six individuals with stroke and three caregivers participated in individual online interviews. An abductive thematic analysis was employed to analyze the qualitative data, using the Unified Theory of Acceptance and Use of Technology 2 (UTAUT-2) model. Results: Participants reported positive experiences with TR, resulting in improvements in functional abilities, such as manual dexterity, balance, and positive interactions with therapists. They found the technology easy to learn and use, facilitating engagement in TR. Participants’ prior experiences with technology, along with support from caregivers and therapists, facilitated acceptance and the use of TR. The COVID-19 pandemic also motivated participants to accept TR. However, technical issues, unstable internet connections, and lack of feedback were barriers to the use of TR. Conclusion: Despite existing obstacles, TR can be used to provide rehabilitation services for individuals with stroke. Addressing these barriers is necessary to promote the widespread and effective use of TR in the context of stroke recovery.

## 1. Introduction

Stroke impacts millions of people worldwide annually [1], resulting in residual effects that, when interacting with the physical and social environment, can lead to restrictions in social participation [2]. Rehabilitation aims to optimize functional recovery and social participation [3]. A supportive environment, including easy access to rehabilitation care, home modifications to facilitate mobility, and emotional support from caregivers, can significantly enhance the chances of recovery [3]. In Canada, rehabilitation services can be provided in various settings, including hospitals and homes [3]. Evidence has shown that home care is associated with significant improvements in participation in activities of daily living compared to hospital care [4]. Indeed, individuals receiving rehabilitation care in hospital settings are inactive for three-fifths of their waking hours, which is not in concordance with the recommendation for the intensity of treatment and stimulation for post-stroke rehabilitation [4]. On the other hand, home care allows for more opportunities for training in functional tasks in individuals with stroke’s natural environment, facilitating their return to performing daily life activities [4].

The Early Supported Discharge (ESD) program consists of offering home-based intensive rehabilitation services to people who have had a stroke and are discharged once medically stabilized [3,5]. It has the potential to reduce costs and is recommended as a solution to optimize functional recovery in individuals with stroke [3] since past studies have shown that ESD can lead to improvement comparable to inpatient rehabilitation for individuals with stroke [6,7]. For instance, a systematic review with meta-analysis of 17 randomized controlled trials, including 2422 individuals with stroke with moderate disability, demonstrated that ESD programs were associated with decreased morbidity and limitations, as well as greater improvement in performance in activities of daily living and higher satisfaction at the end of the scheduled follow-up after six months in comparison to conventional care [7]. Moreover, numerous studies have indicated that patients generally prefer to be discharged from the hospital earlier and to receive rehabilitation at home [8,9,10]. However, various obstacles related to environmental conditions can hinder the ESD rehabilitation professionals’ ability to provide quality services that are in line with guidelines, including intensity of treatment, to persons with stroke in their homes [9,10]. For example, costs associated with frequent travel may act as barriers for rehabilitation professionals to provide home care [9,10]. Additionally, safety and accident risks, such as fall hazards due to weather conditions like snowstorms or rain, can pose additional obstacles for rehabilitation professionals [9,10]. Furthermore, even though the risk of disease transmission is lower at home compared to in a hospital, there is still the potential for contagion when rehabilitation professionals visit individuals with stroke at their residence [11,12]. Therefore, since the COVID-19 pandemic, telerehabilitation (TR) has been increasingly offered to allow rehabilitation professionals to deliver effective and safe home care that is acceptable for them as well as for individuals with stroke and their caregivers [11,12].

TR uses communication technologies, such as wearable devices, the internet, virtual reality, tablets, and phones, to remotely provide rehabilitation services to individuals in their homes and, as such, could be a complementary strategy to traditional in-person rehabilitation [13]. Past studies indicated that TR could lead to improvement similar to that of traditional rehabilitation approaches for individuals with stroke regarding performance in the activities of daily living and quality of life [13,14,15,16,17]. TR has shown to be applicable by rehabilitation professionals, and individuals with stroke perceived that TR was effective [11,15]. Furthermore, it has been shown that individuals residing in rural or remote areas often face difficulties in accessing specialized rehabilitation services that are not locally available [11]. Therefore, through TR, these individuals could benefit from these services, which is particularly advantageous in a vast country like Canada [11,13]. As a result, TR could favor equity between individuals with stroke regardless of their location, which is in concordance with the World Health Organization’s goal of ensuring equal access to high-quality healthcare services [18].

Although TR was shown to be applicable in previous studies, the scientific literature is limited regarding individuals with stroke and their caregivers’ acceptability of TR [16,19]. Applicability in our study is defined as the extent to which TR can be successfully used or applied by rehabilitation professionals and individuals with stroke [11], while acceptability is defined as the extent to which individuals with stroke, their caregivers, and rehabilitation professionals perceive TR as satisfactory, appropriate, and effective for delivering rehabilitation services remotely [11]. Additionally, using rehabilitation technologies in the home environment can be associated with certain barriers, such as complexity of use, limited access to reliable internet and appropriate equipment, and the loss of tactile and kinesthetic aspects of rehabilitation [20,21,22]. A more complete understanding of individuals with stroke and caregivers’ experiences during their rehabilitation when using TR is integrated into the rehabilitation process is essential to ensure quality care is provided for all.

Moreover, few studies have explored the acceptability of TR when used in the home environment of individuals with stroke as part of a rehabilitation follow-up [16,19]. Furthermore, we recently explored the factors influencing the implementation of TR in an ESD program, involving clinicians and managers, and it revealed that numerous factors motivated clinicians to adopt TR [23]. Considering that the perspective of individuals with stroke and their caregivers is lacking, it is imperative that a study exploring their experiences with TR in an ESD program be conducted to better meet their needs. Thus, in alignment with the World Health Organization’s call for more research to improve TR services for individuals with stroke [24], the objective of this study was to explore the acceptability of TR as part of the ESD program and the factors influencing its adoption in individuals with stroke and caregivers.

## 2. Materials and Methods

### 2.1. Research Design

This study was conducted using a descriptive qualitative design [25] that allowed the data analysis to be representative of participants’ own experiences and perceptions on the research subject [26]. The description of this study is presented in accordance with the Consolidated Criteria for Reporting Qualitative Research (COREQ) checklist [27].

#### 2.1.1. Theoretical Framework

Two theoretical frameworks were used, namely the Consolidated Framework for Implementation Research (CFIR) [28] and the Unified Theory of Acceptance and Use of the Technology 2 (UTAUT-2) model [29]. While the CFIR [28] was used to develop interview guides during data collection, as it combined multiple factors that could impact technology adoption, the UTAUT-2 model [29] was used for data analysis, as it allowed us to better understand the relationship between the factors specifically related to technology acceptance.

The UTAUT-2 model was used to determine the behavioral intention to use technology [29]. It is a modified version of the UTAUT model [30] that extends applicability to diverse individuals, such as users, consumers, and customers [31,32,33]. The UTAUT-2 model comprises seven key constructs, with their definition, as follows: (1) “Performance expectancy” pertains to the perceived benefits of using technology; (2) “Effort expectancy” relates to the perceived simplicity of using technology; (3) “Social influence’’ involves the influence of significant individuals on technology adoption; (4) “Facilitating conditions” involve perceiving available resources and support for behavior; (5) ‘’hedonic motivation’’ refers to the pleasure derived from using a technology; (6) ‘’price value’’ represents the balance between technology benefits and financial cost; and (7) ‘’habit’’ denotes the automaticity of a behavior [29,30]. These constructs are believed to influence the behavioral intention to use technology [30]. Additionally, the model incorporates the moderating effects of age, gender, and experience on the seven key constructs, as well as behavioral intention [30]. The UTAUT-2 model has been applied in various fields related to mobile technologies, including healthcare technologies [34,35,36,37] and TR [19]. To improve the rehabilitation process for individuals with stroke and their caregivers, it is essential to understand the acceptability of TR and the factors that influence its use. The UTAUT-2 model is presented in Figure 1.

#### 2.1.2. Population and Recruitment

Participants in this study were individuals with stroke and their caregivers, as they often play an important role in the support and care of individuals with stroke. Including both individuals with stroke and their caregivers (when applicable) in the study allowed for a more diverse range of perspectives and insights. To be eligible for participation, participants were required to speak French or English, to have completed their ESD follow-up, and to have received TR interventions as part of the rehabilitation services the ESD program provided them within the past year. Individuals with stroke with severe cognitive and/or communication impairment that could have limited their participation in the data collection were excluded from the study. Participants were recruited using purposeful sampling [38] among clients from an urban rehabilitation center in Canada offering ESD as part of its stroke rehabilitation program. The aim was to recruit as many participants as possible from those who received TR services in the ESD program.

#### 2.1.3. Intervention

##### The ESD Program

The ESD program provided treatment to individuals in the subacute stroke phase (typically 10 to 20 days post-stroke) who had mild to moderate deficits. These individuals were discharged home once they were medically stable and received the same level of intensive treatment as in an inpatient rehabilitation program [3]. In this current study, participants underwent the ESD program shortly after their hospital discharge, specifically between 7 and 10 days post-discharge. In the ESD program, clients received home-based intensive rehabilitation from an interdisciplinary team following discharge from acute care. The team consisted of an occupational therapist, physiotherapist, speech-language pathologist, social worker, clinical nurse, and special care counselor. The rehabilitation services were tailored to the specific needs of each client and were offered up to five days a week for a duration of five to six weeks. If necessary, clients could be referred to other disciplines, such as nutrition and psychology, to address additional needs. The components of this program were provided remotely to participants using TR. A coordinator facilitated communication between the interdisciplinary team, individuals with stroke, and their caregivers.

##### Telerehabilitation

Before starting remote rehabilitation, members of the interdisciplinary team visited the participants’ homes to assess the environment and provided recommendations regarding suitable spaces for each TR intervention. The intervention proposed during remote rehabilitation was customized to meet the specific needs of each client and included daily life activities. In this study, TR was conducted using various technologies. Clinicians used their workstations (laptops or desktop computers with external camera and microphone). Individuals with stroke and their caregivers used personal cellphones, computers, or iPads. Zoom software version 5.6.5 (823) was used throughout TR sessions.

#### 2.1.4. Data Collection Procedure

##### Sociodemographic and Clinical Characteristics

Sociodemographic characteristics, such as age, level of education, and clinical data, including the professionals from the interdisciplinary team that were involved in their care, the number of TR session for each discipline, and session duration were collected for each participant using an online survey. Additionally, the duration of participation in the ESD program with TR, the date and type of stroke, and time since stroke were extracted from participants with stroke’ medical file.

##### Acceptability and Influencing Factors of Usability

Semi-structured interviews were held remotely in French and recorded using the Zoom Pro platform. These interviews were conducted by the third author, an experienced interviewer and occupational therapist, and lasted 30 to 45 min. The interviewer received guidance from both the second and the last authors. The interview guide was developed based on the CFIR model [28] and consisted of nine main open-ended questions, each accompanied by two to five probes, and was reviewed by the research team, which included experts in rehabilitation as well as knowledge translation, implementation science, and qualitative research. The main themes addressed in these questions revolved around participants’ experience with TR (e.g., could you describe your experience with TR?), the accessibility of the technology (e.g., how did the process unfold to access the technology you needed for TR?), the technology functionality and complexity (e.g., was the technology relatively simple or rather complex to use?), and the interaction with therapists (e.g., how did you find the interaction with your therapists during TR?). The recordings of all interviews were transcribed verbatim by a research assistant and anonymized using pseudonyms. The verbatim transcripts were uploaded to the QDA Miner software version 6.0 to facilitate organization and coding.

#### 2.1.5. Data Analysis

Sociodemographic and clinical data were analyzed using descriptive statistics (means, standard deviations, frequencies, and proportions).

An abductive thematic analysis, which aimed to transcend inductive and deductive approaches, was conducted [39]. Initially, a predefined coding scheme was developed based on the UTAUT-2 constructs. The interview verbatim transcriptions were coded using the predetermined codes, and where relevant, new codes were defined for extracts that were not represented in the initial codes. Finally, the new codes were reviewed with the research team and represented either as sub-themes related to the UTAUT-2 model or as new themes.

#### 2.1.6. Ethical Considerations

Ethical approval for this study was obtained from the Rehabilitation and Physical Impairment Research Ethics Committee (CER RDP) of the Centre intégré universitaire de santé et de services sociaux du Centre-Sud-de-l’Île-de- Montréal (CCSMTL) (Project #2019-1058, CRIR-1347-0618). All participants provided informed consent before taking part in this study and were free to withdraw at any time.

## 3. Results

### 3.1. Description of Sample

A total of six individuals with stroke and three caregivers participated in the study. The mean age of the individuals with stroke was 67.7 ± 10 years, ranging from 50 to 77 years. The mean age of the caregivers was 61.7 ± 17.9 years, ranging from 42 to 77 years. Half of the participants had a university-level education, and 5/9 of them were males. The duration of TR sessions ranged from 45 min to 1 h. The disciplines in the ESD program (e.g., physiotherapy, occupational therapy) that were offered to individuals with stroke depended on their needs (Table 1), and the disciplines that were offered using only TR are mentioned in Table 1. The participants’ strokes occurred between 2020 and 2021, which was during the COVID-19 pandemic when sanitary restrictions were imposed. Table 1 provides details of the participants characteristics.

### 3.2. Acceptability and Factors Influencing Usage of Telerehabilitation

Six main themes following the UTAUT-2 model were identified (Figure 2), which are presented and described in this section: (1) performance expectancy; (2) effort expectancy; (3) facilitating conditions; (4) social influence; (5) hedonic motivation; and (6) habit. Table 2 summarizes the results in terms of facilitators of and barriers to the acceptance and use of TR.

#### 3.2.1. Performance Expectancy

For this theme, two sub-themes emerged: participants’ perceptions (1) before starting TR and (2) after TR.

##### Participants’ Perception before Starting TR

This sub-theme addressed participants’ confidence in TR.

Participants’ confidence in TR: Before starting TR, most participants had no prior experience in using technology for remote rehabilitation sessions. However, they were confident in achieving their rehabilitation goals. “*Before starting, I didn’t have any apprehension, so I went for it […] Since it was the first time […] But in the end, I didn’t have a bad experience, no. It went well*” (IWS-6).

Some participants were confident due to their previous positive experiences with technology and believed it was feasible. Ultimately, the results exceeded their expectations. “*Oh! I had no reservations. I work with technology […] And I knew it was something that could be used […] There were interventions* via *Zoom, and there were things that made progress*” (IWS-3).

##### Participants’ Perception after TR

This sub-theme addressed three key points: (1) positive perception of TR, (2) quality of interaction with rehabilitation professionals, and (3) perception of enhanced functional abilities.

Positive perception of TR: All participants expressed a positive perception of the technology’s use. Their overall experience was marked by significant benefits. The participants found the technology to be reliable and effective, greatly enhancing communication and collaboration with the stakeholders involved in the study. For instance, a caregiver expressed his perspective: “*There were no technological issues, and it allowed for improved communication with various stakeholders in different ways. It was excellent*” (C3). By providing remote rehabilitation services, TR filled a gap in post-stroke care, granting patients continuous access to healthcare professionals and specialized resources. For an individual with stroke, TR had a positive impact on his rehabilitation by providing quick access to treatments. He said: “*It (TR) allowed for faster and more frequent sessions, resulting in a significant benefit for me*” (IWS-3). TR was a valuable and appreciated experience, enabling them to receive the necessary rehabilitation care conveniently and efficiently. ESD, including when provided by TR, was widely perceived in a positive manner, eliciting positive reactions from participants.

Quality of interaction with rehabilitation professionals: All participants appreciated the interactions with the therapists. Despite being remote, the therapists were able to meet the participants’ expectations. Patience, kindness, and availability were the words used by the participants to describe their appreciation of the therapists and to express their satisfaction with the quality of the interactions. For example, in response to the question, “How did you find the contact with the therapists when TR was used? Did you feel a sense of trust developed?” an individual with stroke expressed: “*I found it great! The ladies were great! Super friendly, kind, young. I liked the contact […] they were available […]. I have absolutely no criticism to make, really*” (IWS-2). A caregiver confirmed this by stating: “*I found all the therapists extremely kind and caring! They wanted to make sure things were going well. They wanted to make sure we were doing well. That’s something very positive that I want to emphasize*” (C2). The TR experience was characterized by a good quality interaction with the therapists, which proved to be positive. However, four participants expressed regret regarding the lack of human contact during their TR sessions. They believed that this absence of contact could lead to negative repercussions on the feedback for certain interventions, particularly the challenges in demonstrating different angles to the camera during physiotherapy and occupational therapy sessions. One individual with stroke articulated this concern: “*In terms of physiotherapy, not having someone walking beside you and making corrections as you go along […] it was less, I felt like there was less potential for feedback in terms of physiotherapy*” (IWS-3).

Improvement in functional abilities: All participants mentioned that they had achieved their rehabilitation goals set at the beginning according to their needs. They all reported a significant improvement in their physical condition after TR. Some participants noted an improvement in their manual dexterity, while others mentioned an improvement in their balance. For example, an individual with stroke expressed his success in regaining hand dexterity: “*It was really targeted. Exactly what I needed. Dexterity, the hand. I really succeeded*” (IWS-2). As for a caregiver, her spouse was unable to climb stairs upon leaving the hospital, but he could do so after TR. She expressed: “*On the first day he left the hospital, it took two people on each side to climb the six steps and enter the house. And in the end, he could go up and down all the steps and take a shower on their own*” (C3). ESD, including when provided by TR, has allowed participants to regain their functional abilities in a more effective and efficient manner.

#### 3.2.2. Effort Expectancy

This theme referred to the degree of ease associated with using the system. Two sub-themes related to the technology easy to learn or easy to use and technical issues were identified.

Technology easy to learn or easy to use: All participants expressed their satisfaction with the ease of learning and using the technology. Even those who were using Zoom for the first time found it very easy to learn and use, requiring minimal effort and being well-suited to their abilities. For instance, an individual with stroke stated: “*It was the first time I did it, and it’s easy*” (IWS-2). A caregiver confirmed these remarks, adding, “*It was the first time. We were wondering how it would go. But in the end, it was quite simple*” (C1). The technological tool, Zoom, has proven to be a solution in the context of TR, offering ease of learning and use.

Technical issues: Five participants mentioned technical difficulties, primarily related to their own devices (tablets, laptops, cameras). These difficulties primarily involved audio configuration issues, moving devices in different spaces, camera angles sometimes required for physiotherapy and occupational therapy intervention, and general lighting. For instance, an individual with stroke mentioned issues with the configuration of his tablet, which led to sound problems. He stated: “*I hadn’t properly set up my tablet. So, my tablet had sound and other issues, that were technical problems*” (IWS-6).

Additionally, the stability of the internet connection was mentioned by three participants as one of the technical issues. The quality and smoothness of the audio and visual transmission during TR sessions were influenced by the reliability of the internet connection. However, this technical problem did not hinder the TR sessions, as users generally had a sufficiently good connection to conduct the sessions. For instance, a caregiver emphasized: “*There may have been an internet connection issue at some point, but I don’t have any significant memory of it, generally, everything went well*” (C2).

Despite the technical constraints encountered during TR sessions, participants demonstrated remarkable perseverance and resilience, enabling them to continue their sessions without impediment. No interruptions were observed in the TR sessions due to these technical issues.

#### 3.2.3. Social Influence

This theme pertains to the participants’ perception of the individuals in their lives thoughts about the use of the technology for their rehabilitation. One sub-theme related to family and caregiver influence was identified.

Influence from family and caregivers: Some participants (n = 3) highlighted social influence when using the TR system. In addition to caregivers, family members played a prominent role in exerting social influence. For instance, for one caregiver, she and her daughter formed a strong team to positively influence the TR sessions of their parent. She expressed this by stating, “*We were also a team at home, my husband and his daughter (my daughter too). Thus, it has been an experience that helps people recover and feel comfortable in a familiar environment*” (C3). These participants emphasized the importance of their caregivers’ influence in their therapy using technology. For example, an individual with stroke highlighted: “*I’m fortunate that my spouse is more knowledgeable about technology than I am*” (IWS-5). Another caregiver, due to his mastery of technological knowledge, indicated that he exerted an influence on the technology adoption of his spouse. He stated, “*Fortunately, my own computer knowledge made it happen easily. Because if it had depended solely on my spouse, she wouldn’t have understood anything*” (C2).

Being able to receive support from their caregivers and family facilitated their therapy using technology.

#### 3.2.4. Facilitating Conditions

This theme focuses on the participants’ perceptions regarding the resources and support that were accessible to them during their use of TR. Three sub-themes emerged: (1) in-time support from therapists; (2) equipment used and rehabilitation setting; and (3) physical space and lack of equipment.

In-time support received from therapists: All participants emphasized that they had access to therapist support whenever needed to address technical issues or overcome any other difficulties that could hinder the smooth progress of their TR sessions. This facilitated the tasks during TR sessions. For instance, a caregiver mentioned, “*The first time, she (therapist) always arrived 5 min early to ensure she could assist us with the connection*” (C2). The involvement and availability of the rehabilitation professionals were essential in alleviating concerns and frustrations related to the use of technology, enabling participants to fully focus on their remote rehabilitation process.

Equipment used and rehabilitation setting: All participants used their own equipment, such as computers, iPads, and cameras, during the rehabilitation sessions. They were familiar with these tools and therefore did not need to learn how to use another device, which could have caused additional stress. For example, an individual with stroke used his own tools that he usually used in his professional activities. He stated, “*I have a tablet, a computer. And I used to own a small business that I left maybe a year ago, to retire. So, we were really into technology, we were working with a lot of tools*” (IWS-1).

Furthermore, the fact that the rehabilitation sessions took place at home allowed the participants to feel comfortable and relaxed, eliminating the stress associated with an unfamiliar environment. An individual with stroke expressed his comfort at home by saying, “*It took away the stress of being in a different environment, it puts me in my own environment. There, I could rest as I wanted, I could... I was at home! it says, ‘There’s no place like home’*” (IWS-6). The use of a familiar environment and familiar technological tools by participants undoubtedly facilitated TR by providing them with comfort and autonomy.

Physical space and lack of equipment: Most participants (n = 6) mentioned the need for ample space to perform certain interventions, particularly in occupational therapy and physiotherapy. Some had to rearrange furniture, while others had to move to different rooms depending on the intervention. For example, an individual with stroke highlighted: “*The kitchen is too small, so we used the dining room*” (IWS-4).

Despite these spatial constraints, all participants, in collaboration with the rehabilitation professionals, demonstrated great resilience and managed to find solutions to successfully carry out the TR sessions. One caregiver stated, “*She (rehabilitation professional) told us, look, we’ll do this exercise in that room, and we can do this exercise in this area. To assess things, that’s what she did—she looked at how our home was arranged. Then she conducted tests with certain interventions. And she said, okay, it works here, it doesn’t work here because we need more space, and here it’s safe*” (C1). In conclusion, the limitations arising from space constraints did not prove to be a significant obstacle impeding the smooth progress of TR.

Moreover, some participants (n = 4) highlighted the lack of equipment as an obstacle that can make TR more challenging. For instance, according to one participant, having basic equipment is necessary to engage in TR: “*Nothing too specific, but it definitely requires a foundation. If you don’t have a laptop, it means you’re confined to a single space*” (IWS-5). However, the participants managed to overcome this issue by acquiring the necessary equipment for their TR. For example, an individual with stroke said: “*I had to find my own tools and accessories*”.

#### 3.2.5. Hedonic Motivation

This theme focuses on the pleasure or entertainment derived from the use of TR. Two sub-themes were found related to: (1) society and the health care system and (2) remote accessibility emerged.

##### Society and the Health Care System

This sub-theme addressed the point related to the COVID-19 pandemic.

COVID-19 pandemic: Participants benefited from this intervention (ESD program in the form of TR) during a critical period between 2020 and 2021, while the COVID-19 pandemic was raging. The COVID-19 pandemic and its multiple repercussions on the healthcare system have led to increased difficulty in accessing healthcare services in Quebec. “*We talk about a pandemic because it was during the pandemic […] I couldn’t go to the clinics because they refused due to the virus.*” (IWS-2). Participants quickly realized that TR represented a better option for receiving care, despite the challenging healthcare context, and were motivated to use it. For instance, an individual with stroke did not hesitate to accept TR because of COVID-19, stating: “*I was asked if I wanted to do it* via *Zoom. Given the COVID situation, I said yes*” (IWS-1). Some participants perceived TR as a relief, “*For me, it was a good way to... I wouldn’t leave the house; I wouldn’t have someone come over. I didn’t have access to anyone. We could do exercises and joke around, so it was lighthearted.*” (IWS-5), while others adopted it out of necessity, due to the lack of other available choices, “*We talk about a pandemic because it was during the pandemic. I didn’t have a choice* “(IWS-2).

##### Remote Accessibility

This sub-theme addressed two key points: (1) no need to travel and (2) time-saving.

No need to travel: Most participants (n = 6) emphasized that the absence of travel for both individuals with stroke and rehabilitation professionals made TR more convenient and suitable. For instance, a caregiver emphasized that TR, by eliminating travel, is advantageous for all stakeholders: “*there are benefits to using this technology for the benefit of everyone in terms of travel*” (C3). Furthermore, for one of the participants, the use of TR proved to be an opportune strategy to reduce the risks of virus transmission: “*And with the times (COVID-19 and all), she (the therapist) is safe, and so am I*” (IWS-6). The absence of travel in the context of TR motivated the participants by providing them with a practical solution that is tailored to their specific needs, reduces virus transmission, and overcomes geographical barriers.

Time saving: According to some participants (n = 4), TR allowed them to save a considerable amount of time. For example, for an individual with stroke, TR reduced waiting times and provided a more efficient and convenient means of accessing a diverse range of rehabilitation services within the same day. He expressed it in the following way: “*The virtual approach brings certain advantages, as you can consult multiple specialists in a single day without wasting time on the road*” (IWS-5). TR provided flexibility and allowed individuals with stroke to see a greater number of specialists per day. As perceived by participants, these advantages contributed to optimizing time utilization and improved access to care.

#### 3.2.6. Habit

This theme focuses on the extent to which people tend to engage in behaviors automatically. Two sub-themes related to (1) experience with zoom or other technology and (2) the familiarity with technology related to COVID-19 emerged.

Experience with Zoom or other technologies: All participants had previously used technological tools such as Skype, FaceTime, and even Zoom. Their prior experience boosted their confidence in virtual communication skills, making them more comfortable and willing to use TR. For example, an individual with stroke testified, saying, “*I didn’t really have any doubts because I had already used Skype or similar tools*” (IWS-3). One of the caregivers added, “*We are professionals, and even before COVID, we worked with Zoom, teamwork, or meetings, things like that. We are very familiar with technology. And my spouse (IWS-4) too*” (C3). Overall, the participants’ prior technological experiences played a positive role in their acceptance and successful utilization of TR and had a significant impact on mitigating the influence of advanced age on TR acceptance. Although, some participants (n = 2) pointed out that advanced age, coupled with post-stroke complications like reduced manual dexterity and visual or auditory impairments, could present obstacles to the adoption of technologies, “*For me, everything was complex […], an elderly person who is not accustomed to technology […] I think it would be challenging*” (IWS-5). The other participants did not mention advanced age as an obstacle, possibly due to their previous experience with technology.

Familiarity with technology related to COVID-19: The COVID-19 pandemic significantly expanded participants’ relationships with technology. Most participants (n = 6) gained familiarity with technological tools such as Zoom during the pandemic and continue to use them regularly. One of the participants stated, “*In the context of my work, I used Skype a lot before Zoom came along […]. It was here in Canada that I discovered Zoom because it’s brand new*” (IWS-6). Over the past few years, technology has gradually become an integral part of lives, but the pandemic has accelerated this process dramatically.

#### 3.2.7. Behavioral Intention

Regarding the participants’ behavioral intention to use TR in the future, two sub-themes emerged: (1) participants’ satisfaction with TR and (2) willingness to use TR in the future.

Participants’ satisfaction with TR: All participants were satisfied with TR. They appreciated the opportunity to receive rehabilitation sessions in the comfort of their own homes. Additionally, some participants mentioned that using the technology helped reduce the stress associated with travel and allowed them to adhere to social distancing measures imposed by COVID-19. For example, for caregivers, TR with the ESD program, involving an interdisciplinary team, provided them with great strength. As mentioned by a caregiver, “*So, through the technology […], it was very beneficial for me as a caregiver. I had a whole team around me that gave me a lot of strength*” (C3). An individual with stroke described TR as a positive experience, stating: “*I was very happy to use this system. It relieved me of a lot of stress... It was a quite... let’s say, positive, very positive experience*” (IWS-2). Another participant, an individual with stroke, positively appreciated TR in these words: “*I am amazed and always grateful! So, at no point in any report, I want to see that I have been... No. Everything has been positive, more than positive*” (IWS-6). In conclusion, the participants thoroughly enjoyed the TR.

Willingness to use TR in the future: Overall, all participants reported an improvement in their functional abilities using TR. Consequently, most participants expressed their intention to use the system in the future. For example, for one participant, using TR would pose no problem. She stated, “*So for me, I found that if it had to be done again, I would do it without any problem*” (IWS-2). Similarly, one caregiver affirmed, “*It’s excellent, I wouldn’t change that. There are advantages to using this technology for the benefit of everyone in terms of travel*” (C3).

However, four participants expressed a preference for a hybrid model, combining remote and in-person rehabilitation sessions. For instance, one participant emphasized: “*In my case, since technology is not an issue, I would say it would have been a mix, maybe 90% Zoom and maybe 10% in-person sessions*” (IWS-1). As for caregivers, the hybrid model is seen as necessary: “*Yes, there are advantages to both. Then, I think that if it were used in a hybrid formula for physical rehabilitation, I believe it’s something important to be able to guide more easily. I think that the majority can be done through TR*” (C2).

## 4. Discussion

The objective of this current study was to explore the acceptability of TR and the factors influencing the adoption of TR among individuals with stroke and their caregivers while receiving services from an ESD program. To our knowledge, this study is among the first to have explored the dimensions of TR in an ecological context of the participants and during traditional home-based rehabilitation. The results revealed that various factors related to the majority of UTAUT-2 domains played a significant role in the acceptability and adoption of TR.

First, the findings regarding participants’ perceived performance expectancy revealed that, despite participants’ initial unfamiliarity with TR, they displayed a favorable disposition towards its inclusion/integration in their rehabilitation. Moreover, factors such as improvements in physical abilities, like manual dexterity attributed to TR, and the positive perception of engagement and rapport with therapists positively influenced TR acceptability and its use. These results echoed the conclusions drawn by a qualitative study with 13 individuals with stroke who highlighted improvements in physical abilities and emotional well-being among participants through TR. Another qualitative study with five individuals with total knee arthroplasty conducted by Kairy et al. [40] revealed that participants also appreciated their interactions with physical therapists during their remote rehabilitation sessions, highlighting the importance of a therapeutic relationship for successful rehabilitation [41]. Given these positive results, the majority of participants expressed their satisfaction with TR and indicated their intention to use it in the future, corroborating findings from previous studies where participants similarly reported contentment with TR [40,42,43]. However, some participants reported receiving limited feedback during specific TR interventions, such as physiotherapy and occupational therapy, preferring a hybrid model that combines in-person rehabilitation with a remote component. Previous studies reported similar results, reinforcing the challenges of providing rehabilitation services solely through TR [23,40]. The use of a hybrid model could prove to be a more efficient approach and ensure access to care. Thus, clinicians could choose between face-to-face or TR interventions when appropriate throughout the ESD rehabilitation care episode. However, in countries that do not provide an ESD program, the use of the hybrid model could also help rehabilitation by increasing the frequency of rehabilitation sessions, either individually or potentially in groups. For example, rehabilitation professionals could schedule in-person sessions for comprehensive physical assessments while also offering TR sessions for regular follow-ups. This approach could provide increased flexibility in managing session schedules. Thus, the hybrid model could play a significant role in optimizing rehabilitation care for individuals with stroke, especially in contexts where resources are limited. In summary, these findings demonstrate the acceptability of TR in enhancing the functional abilities of individuals with stroke, as well as the potential to maintain a high-quality therapeutic relationship with healthcare professionals, despite geographical separation, therefore contributing to a favorable performance expectancy for TR and an increased behavioral intention to use it.

Secondly, participants’ perceived effort expectancy underscored that technology-related factors significantly influenced acceptance and the use of TR. Specifically, the Zoom platform used in TR by the ESD program was lauded by participants for its ease of learning and utilization, which has been shown to be a relevant characteristic for the acceptability of TR [42]. However, technical issues and an unstable internet connection presented obstacles, but they did not interrupt the TR sessions. These results could be explained by the fact that the study took place in an urban environment in an industrialized country with excellent internet connectivity or because participants were tolerant of these issues. However, it is worth noting that challenges related to internet access may be even greater in rural areas and/or in developing countries [44]. In comparison to previous studies [23,42], technical problems and internet connection difficulties resulted in visibility issues and poor sound quality during TR sessions. These difficulties led to the cancellation of scheduled sessions and caused subsequent planning problems, which could affect the intensity of therapy for individuals with stroke and could have a negative effect on their rehabilitation [23,42]. Even though individuals with stroke in this current study appeared to be more tolerant of these technical issues, as also noted by a randomized controlled trial with 21 individuals with stroke, conducted by Woolf et al. [45], clinicians would be less so. For example, in their study on the implementation of TR before and during the COVID-19 pandemic, involving six clinicians and two managers, Auger et al. [23] indicated that clinicians had a relatively poor tolerance for technical difficulties and wanted to discontinue TR when these occurred, especially before COVID-19. While technical issues did not disrupt TR for individuals with stroke and caregivers participating in this current study, addressing these concerns should be a priority to ensure a seamless experience for future users. In conclusion, these results emphasize the importance of technology reliability and stability in maintaining the quality and effectiveness of TR programs for individuals with stroke.

Third, participants’ environmental factors (social influence and facilitating conditions) were key for the acceptability of TR, especially regarding caregiver availability, equipment used for the TR, and TR setting. Indeed, individuals with stroke noted that the support provided by their caregivers encouraged them to use and engage with TR more effectively, especially during the early stages of stroke recovery when residual stroke consequences, like fine motor skills problems, visual impairments, and auditory impairments, could add complexity to using the technology. Caregivers played a pivotal role in providing technical assistance, such as device setup and camera adjustment for better visibility, as well as resolving any technical issues, which compensated for individuals with stroke impairments. Additionally, they acted as intermediaries, facilitating communication between individuals with stroke and rehabilitation professionals during TR therapies. Therefore, the presence of caregivers emerged as a significant facilitating factor in the acceptance of TR, aligning with previous studies that confirmed its important and supportive role [43,46,47]. However, not all individuals with stroke in this study had caregivers; yet, they did not report distinct challenges without one, highlighting the acceptance of TR even for those who were alone. Furthermore, participants’ familiarity with their own devices (computers, tablets, cameras) and the TR setting at home facilitated stress reduction and improved the ease of using TR. This aligns with a mixed method study that included seven individuals with stroke, three caregivers, and six clinicians [48] that found that the familiarity of individuals with stroke with the equipment facilitated the adoption of TR. In summary, our evidence illustrates the importance of caregivers being present during TR sessions and equipment familiarity as key factors for successful TR adoption.

Regarding hedonic motivation, the health context related to the COVID-19 pandemic played an important role in motivating the acceptance of TR. Notably, this study focused on individuals who had experienced a stroke between 2020 and 2021, a period that coincided with the height of the COVID-19 pandemic and the corresponding preventive measures, such as social distancing. TR provided these participants with a solution that allowed them to receive rehabilitation care while adhering to social distancing protocols aimed at minimizing the risk of infection. Previous studies have similarly demonstrated that TR served as a practical approach to ensuring continuous care amidst the constraints imposed by the COVID-19 pandemic [49,50,51]. While the initial impact of the pandemic prompted many individuals to adopt TR [52], it also created an environment conducive to widespread familiarity with technologies, particularly platforms like Zoom© and Microsoft Teams© in Canada, and the exploration of their numerous benefits [52]. Therefore, participants’ earlier experience with these technological tools facilitated TR use, aligning with Kruse et al.’s systematic review [53] of 30 studies that highlighted how a lack of technological experience and computer literacy acted as significant barriers to telemedicine adoption. Similarly, Auger et al. [23] observed that, before to the COVID-19 pandemic, clinicians’ and managers’ limited experience with the telemedicine platform imposed on them at that time impeded TR adoption. However, as reported in the same study [23], during the COVID-19 implementation phase, the fact that clinicians and managers themselves chose to use Zoom© because it was easier, and their familiarity with this platform, facilitated the adoption of TR. In summary, COVID-19 prompted participants to recognize certain advantages of TR, such as security, convenience, and accessibility. It would be important for TR to continue playing a significant role in post-stroke rehabilitation to ensure equitable access to rehabilitation services. Future studies could help to better learn from individuals with stroke and caregivers’ experiences to provide essential guidance for clinical practice and policymaking.

## 5. Strengths and Limitations

This study has methodological strengths that enhance the scientific rigor of its findings. First, the use of two robust theoretical frameworks, namely the CFIR [28] and the UTAUT-2 [29] models, significantly bolstered the credibility of this research. Indeed, the incorporation of CFIR [28] enabled us to meticulously construct interview guides, comprehensively probing various facets influencing the adoption of TR within the ESD program. Meanwhile, the application of the UTAUT-2 model [29] during thematic analysis facilitated a profound and exhaustive exploration of the data, aligning with the objectives and unveiling the intricate interplay between identified themes. This deliberate approach amplified the credibility and relevance of this study. In order to ensure the transparency of the methodology, we adhered to the COREQ checklist, reinforcing the transferability of the outcomes. Furthermore, the inclusion of two distinct participant groups, individuals with stroke and caregivers, enriched the diversity of perspectives in this study, thereby reinforcing its credibility.

However, it is important to acknowledge certain limitations inherent in this research. The restricted sample size and homogeneity of participants drawn solely from a single ESD program limited the spectrum of perspectives obtained. This uniformity poses challenges in extrapolating these findings to other post-stroke rehabilitation programs incorporating TR, particularly in varying ESD contexts. However, contextual and personal details were documented and provided in order to allow the transferability of results where relevant. It is also important to acknowledge the presence of selection bias, given that we only had access to a sample of individuals monitored by the ESD program who received TR; this could have led to the collection of generally more positive perceptions. Thus, caution is necessary when attempting to generalize the findings to encompass all individuals with stroke and caregivers. Nevertheless, the meticulous presentation of both methods and results facilitates their adaptability across different settings [54].

## 6. Conclusions

The results of this qualitative study suggest that TR was accepted by individuals with stroke and their caregivers. Perceived benefits, such as improved performance, ease of technology use, and facilitative conditions, including support from caregivers and healthcare professionals, as well as the hedonic motivation associated with the COVID-19 pandemic, were factors that contributed to the acceptability of TR and its adoption among participants. TR is a promising intervention method for stroke rehabilitation by ESD programs, as it has been found to be convenient, accessible, and likely to be adopted in the long term by individuals with stroke and caregivers. However, barriers such as technical issues, unstable internet connections, and lack of feedback could limit the acceptance and use of TR. Thus, the use of a hybrid model combining in-person and remote rehabilitation can help overcome some of the challenges of TR and better address the needs of individuals with stroke and their caregivers. Future research is needed to further deepen our understanding of the needs and responsibilities of individuals with stroke and their caregivers.

## Figures and Tables

**Figure 1 healthcare-12-00365-f001:**
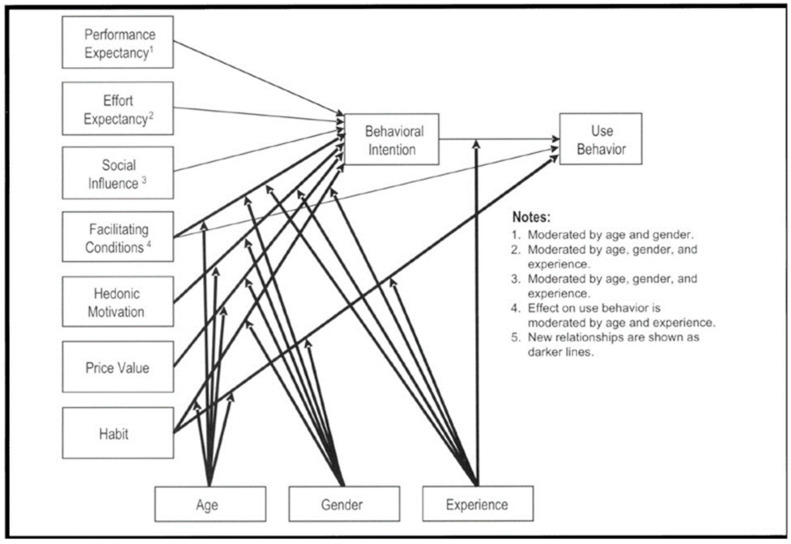
The Unified Theory of Acceptance and Use of Technology 2 (UTAUT-2) model from Venkatesh et al. 2012 [29] (permission to use the figure granted by the author: Viswanath Venkatesh). Note: The lighter arrows in Figure 1 show the original UTAUT domains.

**Figure 2 healthcare-12-00365-f002:**
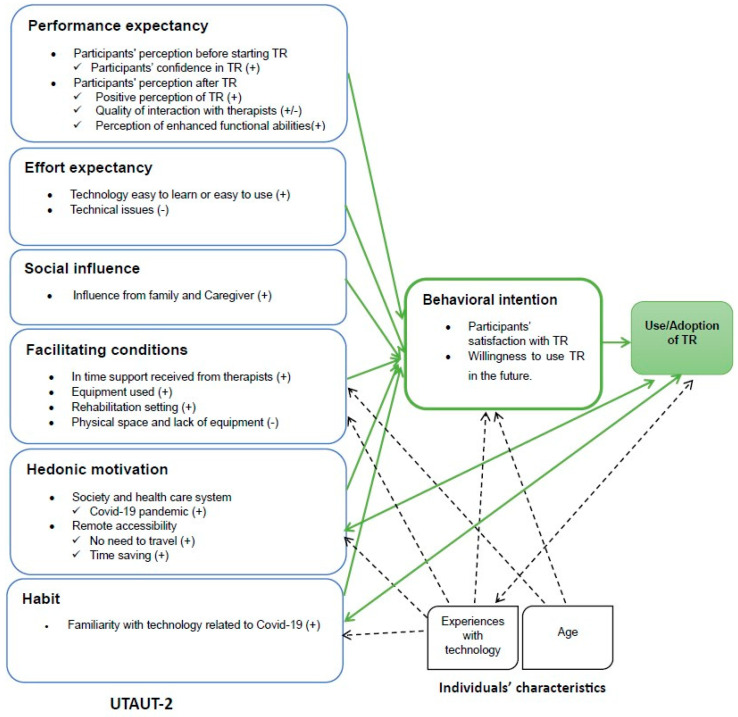
Acceptability and perceived factors influencing the use of TR based on the UTAUT-2 model. Note: + indicates a facilitator; − indicates a barrier; double-sided arrows (
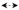
 ↔) show a dynamic, reciprocal relationship between two constructs, indicating influence in both directions. One-way arrows (→ ⇢) depict a unilateral relationship, emphasizing that one construct can impact the other without direct reciprocity.

**Table 1 healthcare-12-00365-t001:** Participants’ IDs and personal characteristics.

Participants ID	Age (Years)	Genre	Level of Education	Type of Stroke	ESD Duration with TR Modality (Weeks)	Rehabilitation Disciplines Involved	Disciplines Using only TR/Number of TR Session/Week	Time Since Stroke at the Time of the Interview (Months)
Individuals With Stroke (IWS)								
IWS-1	76	Male	University	Ischemic	5	PT, OT, SLP, SW; nurse, Psy	- OT/5/week - PT/5/week - SLP/5/week	12
IWS-2	77	Female	University	Ischemic	5	OT, kinesiology	- OT/3/week - Kinesiology/3/week	11
IWS-3	65	Male	Secondary	Ischemic	5	Psy, PT, SLP	- SLP/3/week	12
IWS-4	72	Male	University	Ischemic	6	OT, SLP, SW, psy, PT	- OT/3/week - PT/3/week - SLP/3/week - SW/2/6weeks	12
IWS-5	50	Female	College-level	Hemorrhagic	5	OT, PT, nurse, SW, art therapy	- OT/5/week - PT/5/week - Art therapy/5/week - Nurse/2/5weeks	7
IWS-6	66	Male	University	Ischemic	4	OT, PT, SW	- OT/3/week - SW/3/week	8
Caregivers (C)								
C1 (IWS-5)	42	Male	Secondary	N/A	N/A	N/A	N/A	N/A
C2 (IWS-2)	66	Female	University	N/A	N/A	N/A	N/A	N/A
C3 (IWS-4)	77	Female	Vocational diploma	N/A	N/A	N/A	N/A	N/A

Note: PT: physiotherapy; OT: occupational therapy; SLP: speech language pathology; SW: social worker; Psy: psychology/neuropsychology; N/A: not applicable.

**Table 2 healthcare-12-00365-t002:** Facilitators of and barriers to the acceptance and use of TR.

Factors Facilitating the Acceptance and Use of Telerehabilitation	Barriers to the Acceptance and Use of Telerehabilitation
Participants’ confidence in TR. Positive perception of TR. Quality of interaction with rehabilitation professionals. Improvement in functional abilities. Technology that is easy to learn or easy to use. Support from family and caregivers. In-time support received from therapists. Familiarity with the equipment used for telerehabilitation (their own computer, iPad, tablet). Familiarity with the rehabilitation setting (in their home environment). No need to travel. Time-saving. Prior experience with Zoom or other technologies.	Technical issues (audio configuration issues, relocating devices to different spaces, adjusting camera angles, and addressing general lighting concerns). Instability of the internet connection. Need for ample space to perform certain interventions (physiotherapy and occupational therapy). Lack of equipment. Lack of human contact. Lack of feedback.

## Data Availability

The data are not publicly available due to confidentiality and in order to comply with the requirements of the research ethics committee.

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
