# Peer review of "Acceptability of Telerehabilitation: Experiences and Perceptions by Individuals with Stroke and Caregivers in an Early Supported Discharge Program"

_healthcare, 2024, doi:10.3390/healthcare12030365_

Round 1
Reviewer 1 Report
Comments and Suggestions for Authors
ABSTRACT
1. A distribution should appear in this, respecting the same distribution as the rest of the text (introduction, methods, results, etc.).
2. The results obtained and the conclusion of the study should appear in the abstract.
INTRODUCTION
Furthermore, providing rehabilitation services in an individual with stroke's home could increase the risk of disease transmission for both in- 69 dividuals with stroke and rehabilitation professionals [12]". Is there more risk of disease transmission in a home setting than in a hospital where there are many people with different pathologies? Are you sure about this? Need more references with other studies.
2. The introduction is a bit confusing, is the TR applicable or not? WHO recommends it but then the scientific evidence is limited? This section should be clarified and rewritten, it is not clear what the authors want to express.
3. A randomized controlled trial is not representative? I do not agree. However, a descriptive study such as the one carried out would be more so? The authors need to provide an explanation for this statement.
4. Reference is made to COVID in the last paragraphs, however the current pandemic situation has changed, it does not seem a good argument to justify the need for this study.
5. The objective of the study and the research question are not included in the last paragraph of the introduction, it would be of interest to do so.
METHODS
1. Is there a sample size calculation? Why only that number of patients?
2. The figure seems good to me, however the arrows are some highlighted and others not, is this because of something in particular? If so, it should be explained, if not, I recommend that all of them have the same tonality.
3. Why are the caregivers also included? In which cases? It should be explained and a justification for this should be given.
4. Does the fact that acute and subacute patients are included not affect the results of the study? It would be necessary to justify why both
5. We talk about an observational study but then an intervention is made, it is not coherent.
RESULTS
1. It does not appear in the legend N/A explained.
2. Although the results are well expressed, the comprehension with so much text is not easy, it would be of interest to include a table 2 where these results are visually easier to understand.
REFERENCES
1. They are not in the correct format, please revise them.
Reviewer 2 Report
Comments and Suggestions for Authors
Thank you for the opportunity to review the article.
Stroke and its consequences are a factor that significantly affects the quality of life of patients. Early and continuous rehabilitation significantly improves the functional status of patients after stroke and is also a factor that relieves the burden on caregivers. The work shows aspects of telerehabilitation that may affect the functioning of patients and caregivers in everyday activities
My comments concern:
-
Purposeful selection of the research group, which may affect the external validity of the results obtained;
-
The study group is very small and does not constitute a representative sample for post-stroke patients;
-
The selection was guided by simplicity, ease and factors clearly indicating the selection of respondents for the study;
-
The choice of an experienced interviewer in French is questionable, which clearly deprives people of other languages ​​from participating in the survey;
-
Lack of information about the duration of the TR session and whether the participants' involvement was assessed for correctness;
-
A clear form of discussing the respondents' answers would be to summarize the advantages and disadvantages of telerehabilitation sessions in a table;
